# Effects of Osseodensification on Immediate Implant Placement: Retrospective Analysis of 211 Implants

**DOI:** 10.3390/ma15103539

**Published:** 2022-05-15

**Authors:** Márcio de Carvalho Formiga, Kinga Grzech-Leśniak, Vittorio Moraschini, Jamil Awad Shibli, Rodrigo Neiva

**Affiliations:** 1Department of Periodontology and Oral Implantology, Unisul, Palhoça 88137-270, Brazil; marciocformiga@gmail.com; 2Laser Laboratory Oral Surgery Department, Medical University of Wroclaw, 50-425 Wroclaw, Poland; kgl@periocare.pl; 3Department of Periodontology, Veiga de Almeida University, Rio de Janeiro 20271-020, Brazil; vitt.mf@gmail.com; 4Department of Periodontology and Oral Implantology, Dental Research Division, University of Guarulhos, Guarulhos 07023-040, Brazil; 5Department of Periodontics, School of Dental Medicine, University of Pennsylvania, Philadelphia, PA 19104, USA; rneiva@upenn.edu

**Keywords:** Osseodensification, bone instrumentation, dental implants

## Abstract

Osseodensification is a new method of bone instrumentation for dental implant placement that preserves bulk bone and increases primary implant stability, and may accelerate the implant rehabilitation treatment period and provide higher success and survival rates than conventional methods. The aim of this retrospective study was to evaluate and discuss results obtained on immediate implant placement with immediate and delayed loading protocols under Osseodensification bone instrumentation. This study included private practice patients that required dental implant rehabilitation, between February 2017 and October 2019. All implants were placed under Osseodensification and had to be in function for at least 12 months to be included on the study. A total of 211 implants were included in the study, with a 98.1% total survival rate (97.9% in the maxilla and 98.5% in the mandible). For immediate implants with immediate load, 99.2% survival rate was achieved, and 100% survival rate for immediate implant placement without immediate load cases. A total of four implants were lost during this period, and all of them were lost within two months after placement. Within the limitations of this study, it can be concluded that Osseodensification bone instrumentation provided similar or better results on survival rates than conventional bone instrumentation.

## 1. Introduction

Dental implants have revolutionized methods for oral rehabilitation. Initially used for full-arch mandibular rehabilitation, dental implants are now considered to be the standard of care for single, partial, and full-arch teeth replacement [1,2,3]. Total treatment time has also been significantly reduced with advances in the implants’ macro- and microgeometry, surface treatment, as well as types of implant prosthetic connections and other factors [4,5,6,7,8]. However, bone instrumentation for the placement of dental implants has been largely overlooked in implant studies [9,10,11]. Since the success of implant osseointegration is directly related to implant primary stability, under-sized osteotomies have been used to maximize initial bone to implant contact, especially in areas of low bone density, to increase implant primary stability [12,13,14,15]. However, this approach may affect secondary stability due to excessive bone compression and ischemia [16,17]. In addition, techniques performing osteotomies with piezosurgery may also improve the primary stability of implants. Even with the risk of overheating using ultrasonic devices, it may stimulate the bone during site preparation and enhance osseointegration [18,19]. The use of bone compactors in low density bone may also increase primary stability of dental implants through lateral bone compression [20].

First described by Huwais and Meyer [21] in 2016, the concept of Osseodensification (OD) is based on bone instrumentation with minimal trauma and the creation of an implant osteotomy with increased bone density by autologous lateral bone compaction. Hence, bone to implant contact is increased, resulting in higher insertion torque [21,22,23,24,25,26]. The clinical significance of these findings corresponds to increased success rates in immediate and early loading protocols, resulting in greater patient satisfaction.

Because the Osseodensification technique is based on bone bulk preservation by compaction of cancellous bone due to viscoelastic and plastic deformation, and at the same time compacting autographs of bone chips [21], it has been demonstrated that this increases the bone to implant contact, without bone extraction, and therefore increases implant primary stability [21,22,23,24,25,26]. In low density bone, this advantage may be critical to decrease implants’ micro-motion during osseointegration, and increase success rates. Another claimed advantage of the Osseodensification burs is the reduction of the size of osteotomies when the burs are removed, called the spring-back effect, due to the viscoelastic portion of the deformation [21]. The bone recovery to the center of the osteotomy may also help obtain higher insertion torques and, with that, make more immediate loads viable than with regular bone instrumentation techniques [23,24,25,26].

The aim of this retrospective study was to evaluate the survival rate of 211 dental implants placed with Osseodensification bone instrumentation.

## 2. Materials and Methods

This retrospective study was conducted in compliance with the Declaration of Helsinki and approved by the Ethics Committee of the University of Guarulhos, process # 205/2003, for studies involving humans, and included private practice patients in need of dental implant rehabilitation who searched for dental implant treatments from February 2017 to October 2019. All data were extracted from the same periodontist private office patients. All patients treated during the selected period received the same type of implant, and were instrumented with Osseodensification. Treated cases included maxillary and mandibular implant placement with tapered implants and morse taper connection (Figure 1), in native bone and grafted sites, as well as immediate implant placement, with and without immediate loading. All bone instrumentations were made with the Versah Burs (Figure 2), with abundant irrigation, at 1100 rpm. The drilling protocol followed the suggested protocols by the manufacturer (Versah), i.e., pilot bur on clockwise rotation, 2.0. 2.3, 3.0, 3.3, 4.0, 4.3 and 5.0 burs on counterclockwise rotation on immediate implant placements, accordingly to the length and diameter of the selected implant. The final bur used had a closer diameter within the diameter of the implant, which means undersized instrumentation was not necessary with this method. For an example, if the selected implant was 3.5 × 9 mm, the last bur used was the VT2838(3.3) because only tapered implants will be installed. On type I and II bone (late implant placements), the instrumentation was made on clockwise rotation following the same sequence, but only the last bur was instrumented on counterclockwise rotation. This report focuses on the effects of Osseodensification instrumentation protocols on immediate implant placement, with and without immediate implant loading cases. For immediate implant placement and immediate load for single tooth cases, the sequence was: extraction (Figure 3 and Figure 4), implant drilling (Figure 5), implant placement (Figure 6), abutment selection and installation, gap filling with xenograph (Figure 7), and temporary crown installation (Figure 8). After 4 to 6 months, a metalo-ceramic crown was delivered (Figure 9), and after 6 months a control x-ray was taken to evaluate bone loss (Figure 10). On immediate implant placement without load cases, the sequence was: tooth extraction (Figure 11 and Figure 12), implant drilling (Figure 13), implant placement (Figure 14), gap filling with xenograph (Figure 15), and socket sealing with a dense PTFE barrier (Figure 16). After 4 to 6 months, re-entry occurred, the temporary crown was delivered, and after mucosa conditioning the final crown was delivered (Figure 17). After 6 months in function, a control x-ray was taken (Figure 18). Patients with uncontrolled systemic conditions were excluded from this observation due to their contra-indications for dental implant surgeries. At 6 months in function all patients were called back for a clinical and radiographic evaluation, where all peri-implant mucosa were probed, and a periapical x-ray was taken to evaluate survival criteria as listed above. Implant survival was defined as implants that remained in function for at least 6 months following prosthetic rehabilitation and presented with less than 3 mm of crestal bone remodeling, without pain, implant mobility, or suppuration on probing. All dental implants (Duecone, Implacil de Bortoli- São Paulo, Brazil) were instrumented with Osseodensification and followed the protocols suggested by the Osseodensification bur manufacturer ( Versah LLC, Jackson, MI, USA). Because Osseodensification is indicated where there is a minimum of 2 mm of cancellous bone [21], whereas the bone type was I or II, the burs should work on clockwise rotation, and only the last one or two final drills on counter clockwise rotation. Moreover, whereas the bone was type III or IV, Osseodensification could be performed since bur 2.0. A xenograft (Lumina Porous Small, Criteria, São Carlos, SP, Brazil) was used to fill the gaps between the implants and the socket walls when implants were immediately loaded.

All dense polytetrafluorethylene (PTFE) barriers were removed after 21–28 days. Statistical analysis of the implant survival rates was performed with life-table analysis as previously described [27].

## 3. Results

A total of 211 implants were placed by means of Osseodensification protocols in 132 subjects (71 males, 61 females) and remained in function for at least twelve months to be included in this retrospective evaluation. Mean age of 52 years (Table 1). Because the present study is a retrospective study of a specific periodontist private practice, and so it was not a designed study from the start, basically the only inclusion criterion was: patients who underwent implant treatment by this periodontist from the beginning of his experience with the Versah burs (Osseodensification drills). The same occurred with the exclusion criteria: since all implants placed in the selected period were instrumented with Osseodensification burs, there were no exclusions to be analyzed. Exclusions were only to occur if a patient abandoned the treatment, which did not happen. The total survival rate was 98.1% for the implants inserted by Osseodensification, of which 97.9% were in the maxilla, and 98.5% in the mandible (Table 2). Four implants were lost: one following immediate implant placement in molar site; one loss during reopening; and two at the two-week post-op appointment (Table 3). All placed implants presented insertion torque ≥40 Ncm despite implant size or location in the mouth.

In cases of immediate placement with immediate load, the success rate observed was 99.2%. One implant was lost within two weeks post-operative when immediate placement was performed on a maxillary molar site with immediate loading. No failures were observed when the immediate implant was not immediately loaded. With respect to implant diameters on immediate loading, a 100% success rate was observed for narrow implants, 98.6% for regular diameter implants, and 100% for wide diameter implants.

Smoking history did not seem to influence implant success since none of the failures occurred in smokers. This observation included 27 smokers: 18 subjects were classified as light smokers (up to 10 cigarettes per day) and the remaining nine subjects were heavy smokers (over 10 cigarettes per day).

All failures occurred within the first 12 months of application of Osseodensification protocols. No additional implant failures were observed after this period. Implants included remained in function from 12 to 32 months (mean: 25 months). Neither age nor gender seem to have influenced the results.

## 4. Discussion

The recently introduced concept of Osseodensification has limited implant survival rates in humans, as reported in the literature [28,29]. One study reported a survival rate of 97% in crestal maxillary sinus lift procedures [29], which was similar to the success rate observed in this study (98.1%). Another study reported survival of 92.8% when Osseodensification was used for alveolar ridge expansion [28], which was inferior to the 98.1% success rate in this observation. A recent study showed a 100% success rate on implants placed by Osseodensification, but with a small sample size (10 implants) [30]. This report provides further evidence to support Osseodensification as a valid method to increase success rates in immediate implant placement protocols, with and without immediate loading, using a larger sample size. Despite the limited long-term evidence of success, Osseodensification has shown to improve primary stability of dental implants [23,24,26,31]. This appears to be more significant during immediate loading protocols, where high insertion torques are required for successful treatment. Recent in vitro studies compared standard drilling sequences with Osseodensification protocols in low-density polyurethane blocks, and also concluded that OD resulted in higher primary stability values [32,33]. Fresh extraction sites are known to provide reduced insertion torque and consequently inferior primary stability for implant placement. A recent study, however, presented a 93.1% implant survival rate in immediate implant placement in molar areas with septum expansion instrumented by Osseodensification [34]. In our study, all placed implants presented insertion torque ≥40 N/cm despite implant size or location in the mouth. This is a highly significant finding that provides additional clinical evidence that Osseodensification increases clinician confidence and predictability during immediate implant placement. Clinician skill is an important factor that can influence the level of primary stability of the implants in different techniques [35], and must be similar to Osseodensification burs. All implants analyzed on this study were placed by a periodontist with more than 18 years of implant surgery experience.

The posterior maxilla is known to be the area in the oral cavity with the lowest bone density and with inferior implant insertion torque values. This reduced bone density results in more conservative loading protocols and higher failure rates with conventional osteotomy preparation techniques. Peron and Romanos (2020) [36] reported a survival rate of 100% with immediate loading in the posterior maxilla, identical to the rate found in this study in healed sites in the posterior maxilla, and the immediate load with narrow and wide implants. However, a recent systematic review and meta-analysis [37] showed only 94.9% implant survival in immediate implant placements, versus 98.9% in delayed implant placement cases. With 99.2% success in immediate implant with immediate load cases, it can be suggested that Osseodensification may have improved the clinical results due to the higher primary stability achieved [24].

All implants used in this study were tapered, with morse tapper connection, and with acid-etched surface treatment. These characteristics had been already studied and proven to achieve high success rates [38,39,40,41] in dental implant treatments.

Smoking is considered a risk factor for successful osseointegration of dental implants, as well as for long-term implant success [42]. Smoking did not have a negative influence on the results obtained in these patients. None of the patients who reported smoking habits lost implants or presented complications. This was probably due to less traumatic bone instrumentation with Osseodensification protocols. This is in agreement with Alsahhaf et al. (2019) [43], who reported similar findings in a five-year follow-up study.

The only mandibular implant failure observed in this retrospective study may have been due to higher bone density observed in the mandible without adequate trabecular bone pattern for Osseodensification. Almutary et al. (2018) [44] demonstrated that Osseodensification may not be effective in cortical bone, and may function differently from trabecular bone by decreasing bone repair and delaying or preventing osseointegration, in spite of increased primary stability of dental implants. The need for at least 2 mm of trabecular bone in order to be performed could be considered a limitation of the Osseodensification technique, and because of this, it may not be as effective in type I bone as in types III or IV [21].

This retrospective study presented some limitations, such as the absence of primary stability measurement using ISQ data and different ranges of lengths and diameters. However, when we analyze Table 2, we can see that results for narrow and wide implants were similar. In addition, when we compared survival rates on the mandible and maxilla, we did not find clinically significant differences.

## 5. Conclusions

Based on the limitation of this retrospective study it can be concluded that Osseodensification provides similar or even superior success rates when compared to success rates reported in the literature of conventional methods of bone instrumentation. The increased primary stability obtained with this method of osteotomy preparation does not seem to have a negative impact on implant success. More controlled studies with longer observation periods are needed to support these findings.

## Figures and Tables

**Figure 1 materials-15-03539-f001:**
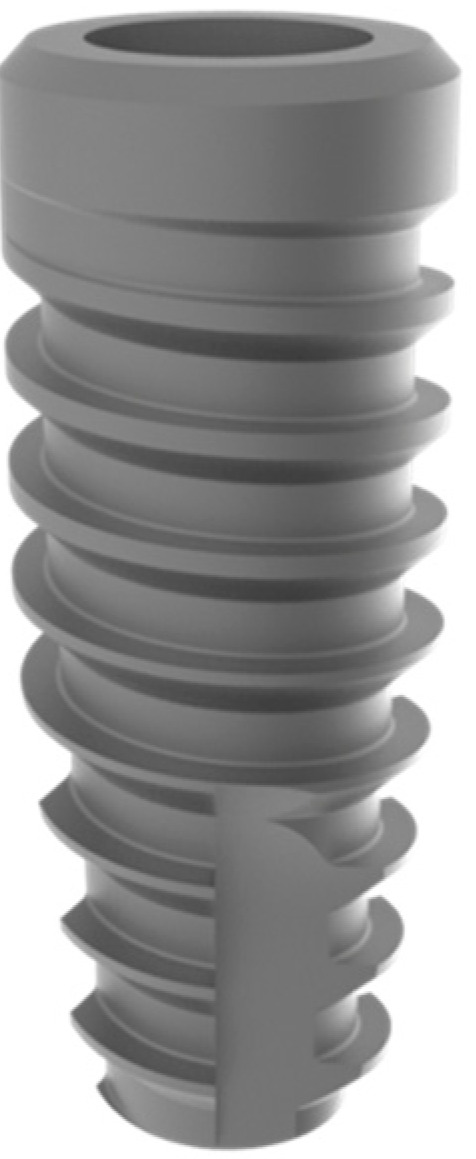
Dental Implant with morse tapper connection (Implacil de Bortoli, São Paulo, SP-Brazil).

**Figure 2 materials-15-03539-f002:**
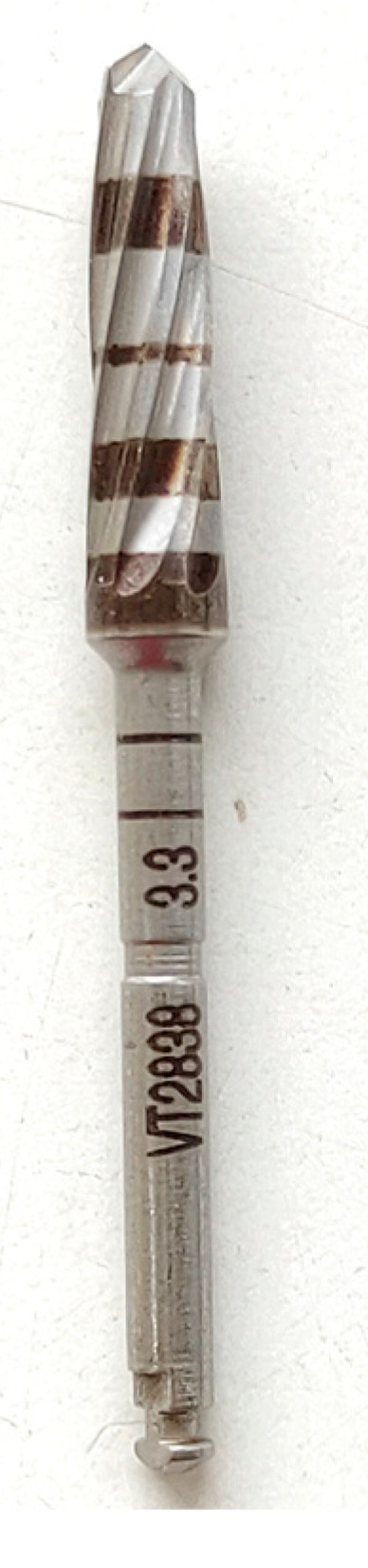
Osseodensification drill VT2838(3.3) (Versah LLC).

**Figure 3 materials-15-03539-f003:**
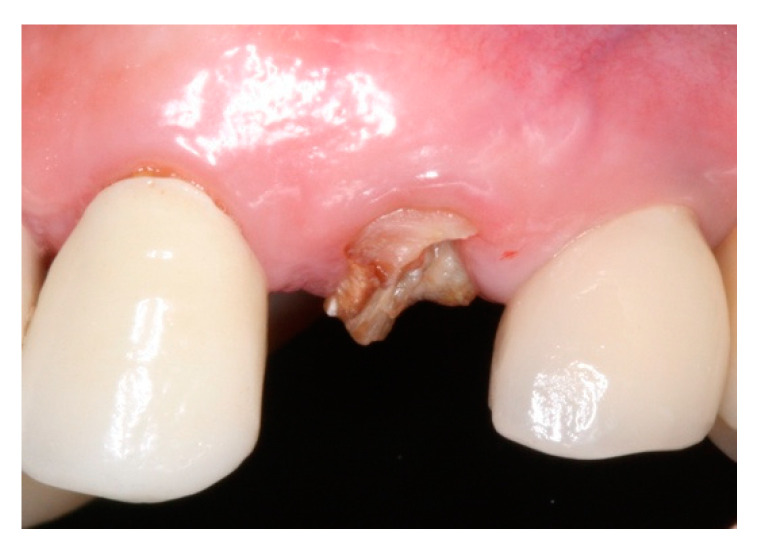
Intraoral buccal view of #24, with a limited dental structure after fractured crown.

**Figure 4 materials-15-03539-f004:**
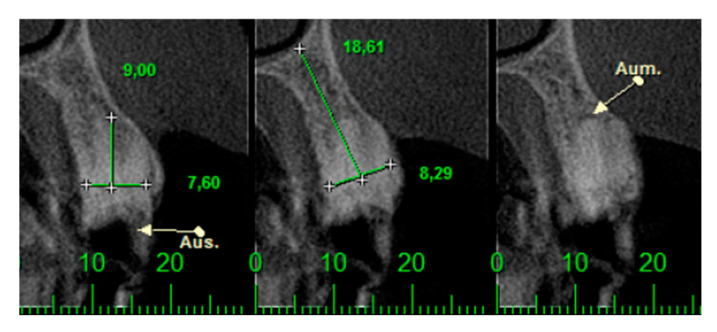
CTCB images of #24 and the radicular fragment.

**Figure 5 materials-15-03539-f005:**
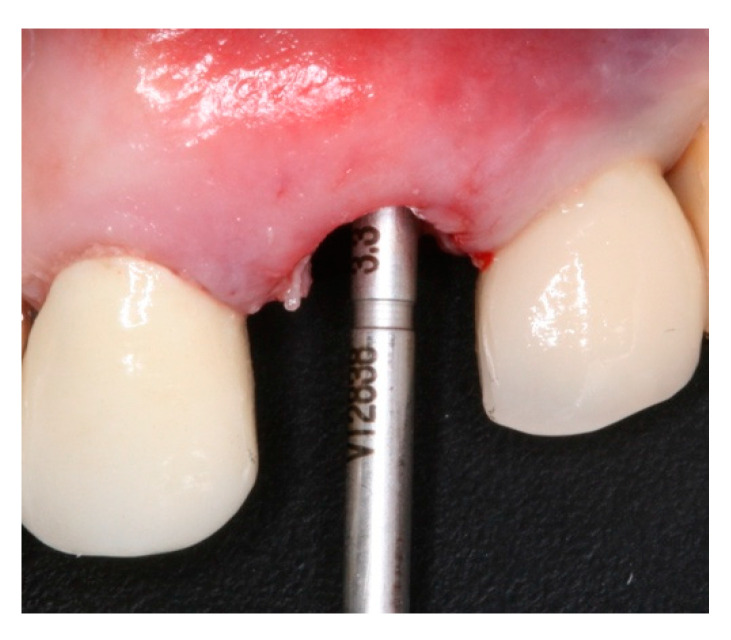
Intraoral buccal view of Versah drill VT2838(3.3) on #24 after site instrumentation.

**Figure 6 materials-15-03539-f006:**
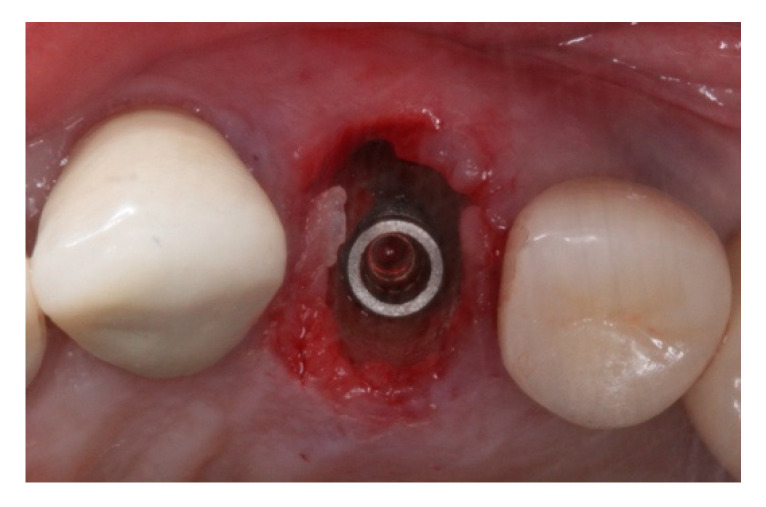
Intraoral oclusal view of morse taper implant placed on #24 site after Osseodensification bone instrumentation.

**Figure 7 materials-15-03539-f007:**
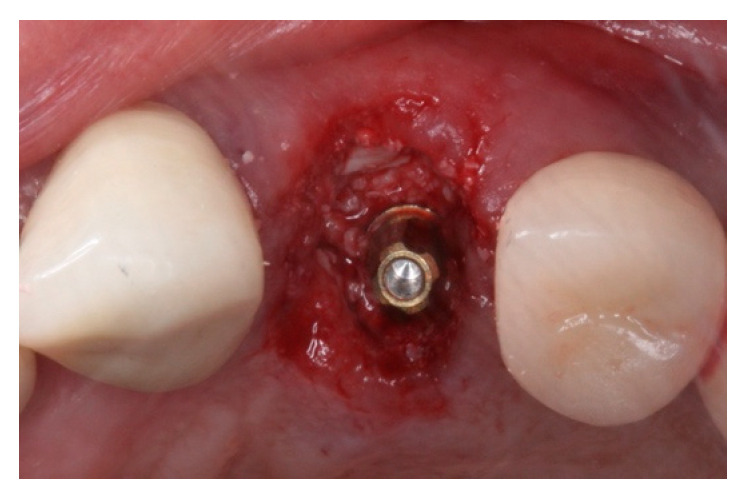
Immediate implant grafted with xenograft material. Note that the prosthetic abutment was installed to receive the interim prosthesis.

**Figure 8 materials-15-03539-f008:**
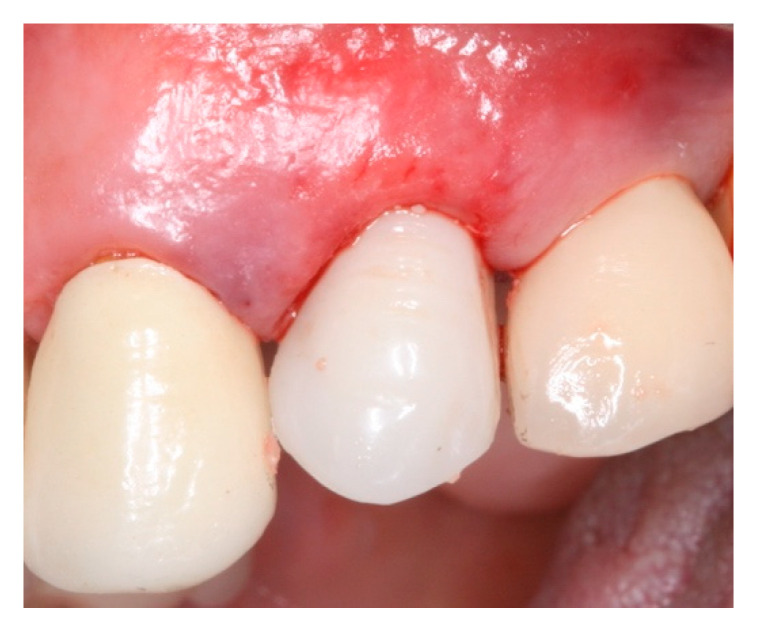
Interim prosthesis (screw retained restoration) on immediate implant placed on #24.

**Figure 9 materials-15-03539-f009:**
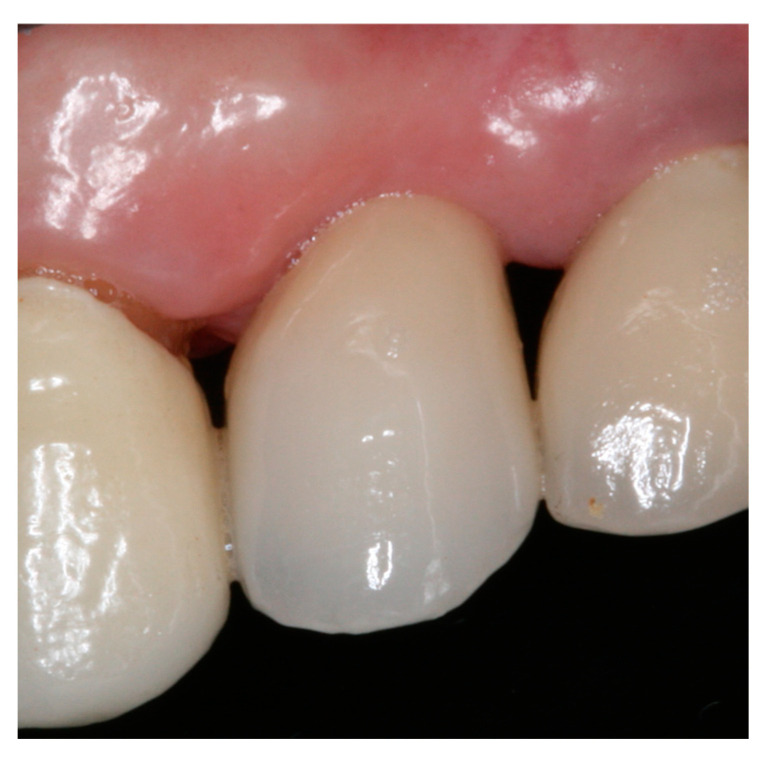
Intra-oral buccal view of #24 with ceramic screw-retained crown after 14 months in function.

**Figure 10 materials-15-03539-f010:**
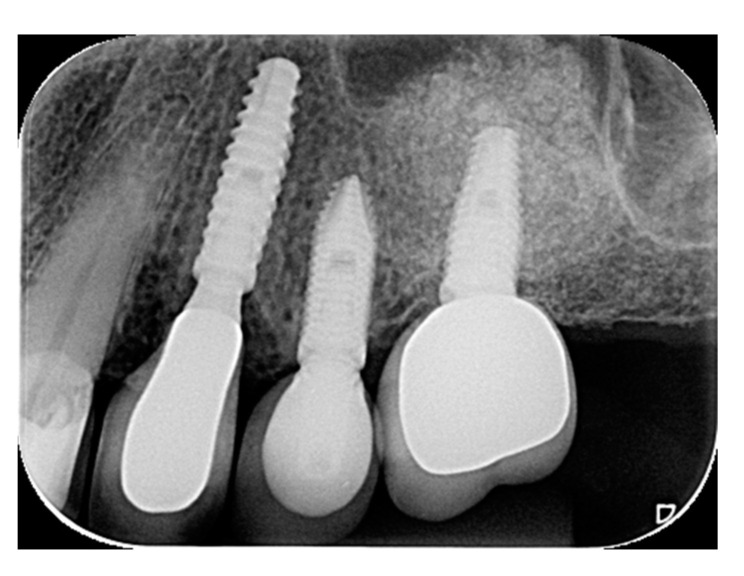
Periapical radiograph of #24 in function for 14 months.

**Figure 11 materials-15-03539-f011:**
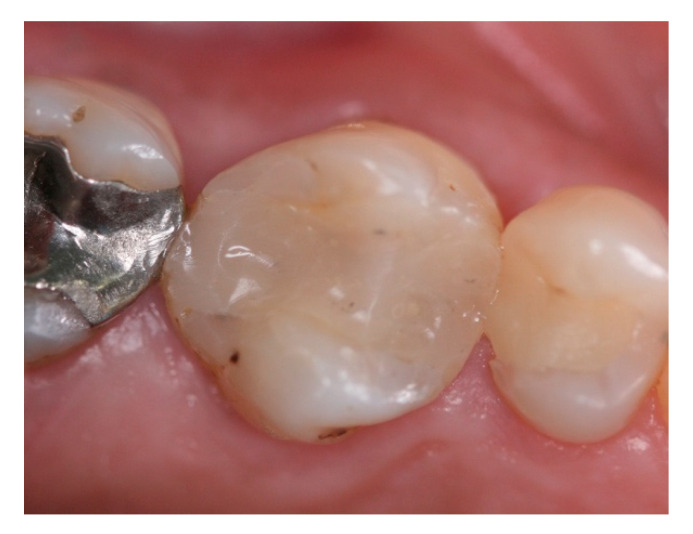
Intra-oral oclusal view of #16, indicated for extraction.

**Figure 12 materials-15-03539-f012:**
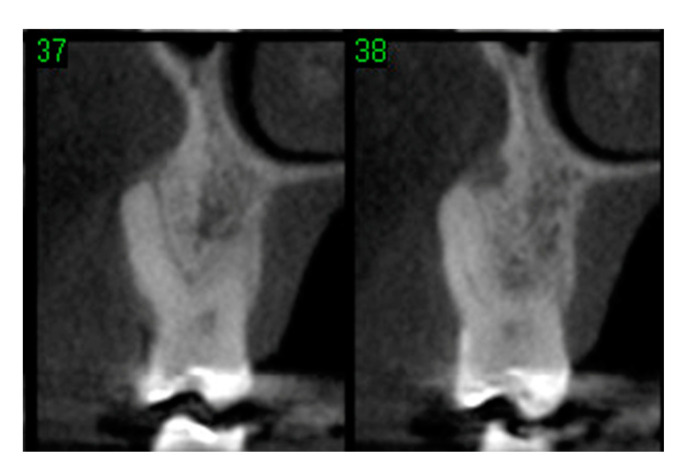
Computerized tomographic view of 2 slices (37 and 38) of #16 evidencing lack of buccal plate.

**Figure 13 materials-15-03539-f013:**
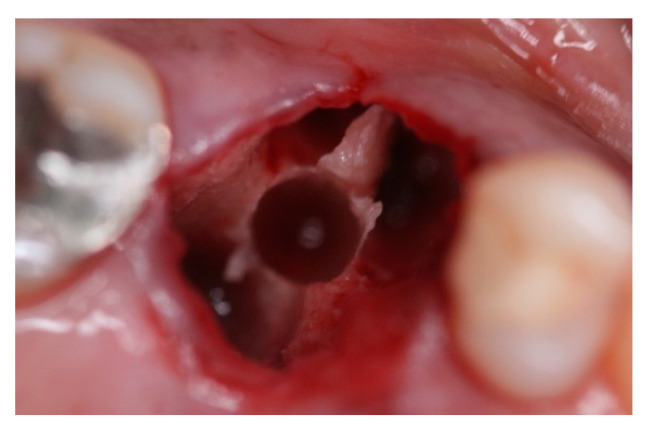
Intra-oral oclusal view of #16 site after extraction and septum expansion by Osseodensification bone instrumentation.

**Figure 14 materials-15-03539-f014:**
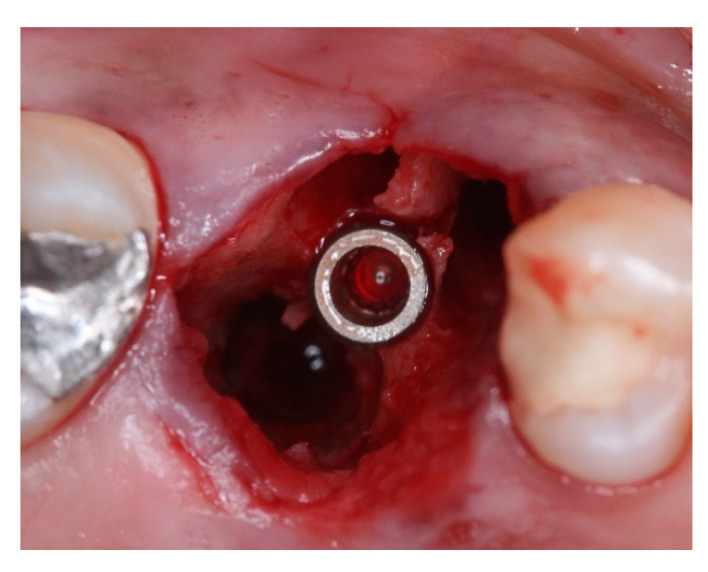
Intra-oral oclusal view of #16 after morse tapper implant placement.

**Figure 15 materials-15-03539-f015:**
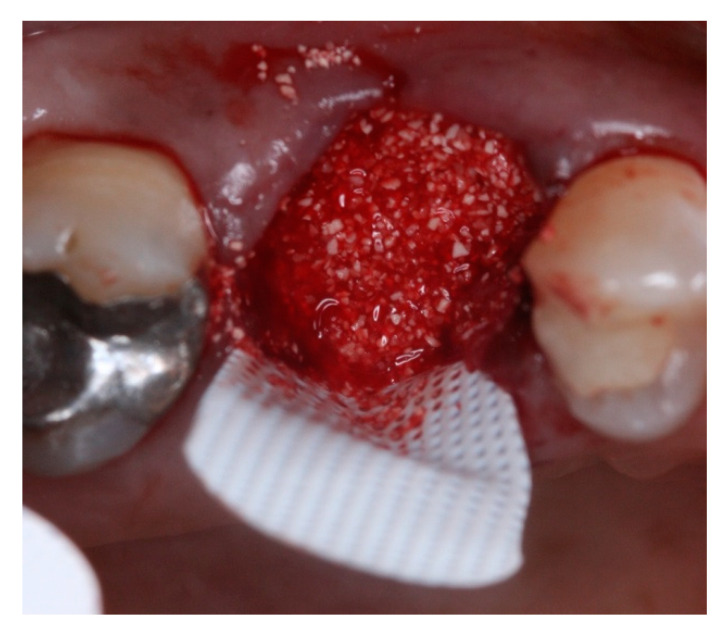
Intra-oral oclusal view of #16 after gaps filled with xenograft.

**Figure 16 materials-15-03539-f016:**
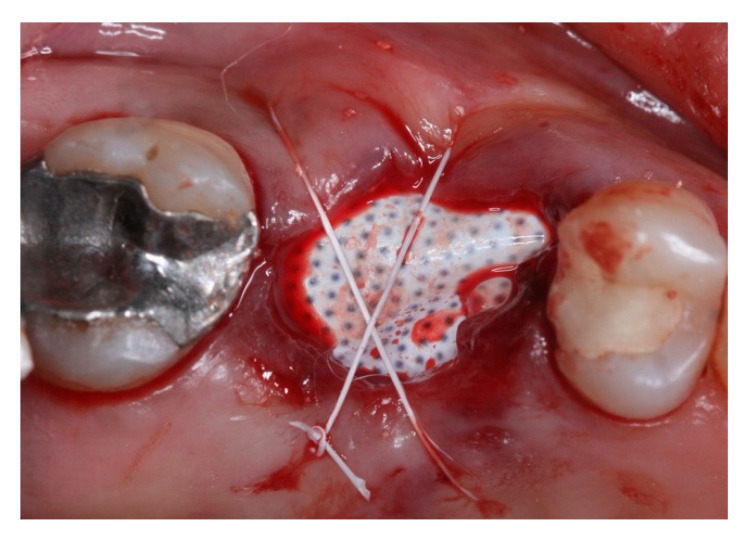
Intra-oral oclusal view of #16 with dense PTFE barrier installed to seal the alveoli and sutured with PTFE thread.

**Figure 17 materials-15-03539-f017:**
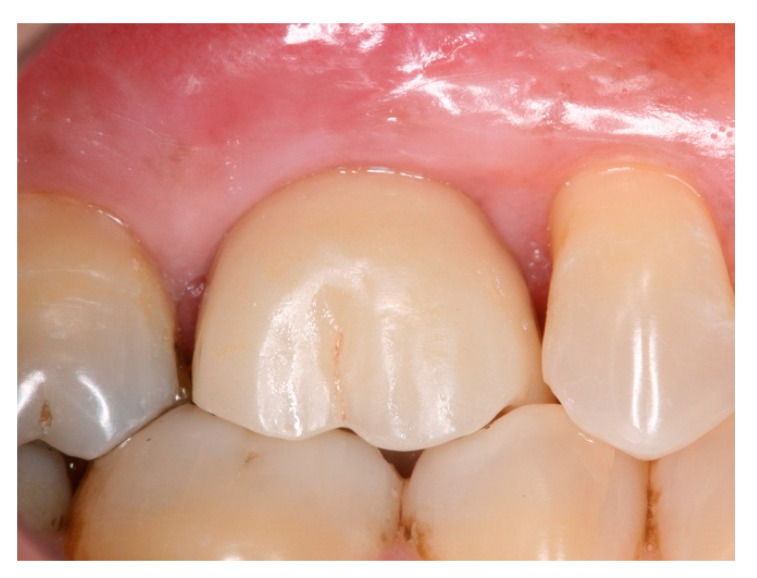
Intra-oral buccal view of #16 with ceramic screwed retained crown in function for 25 months.

**Figure 18 materials-15-03539-f018:**
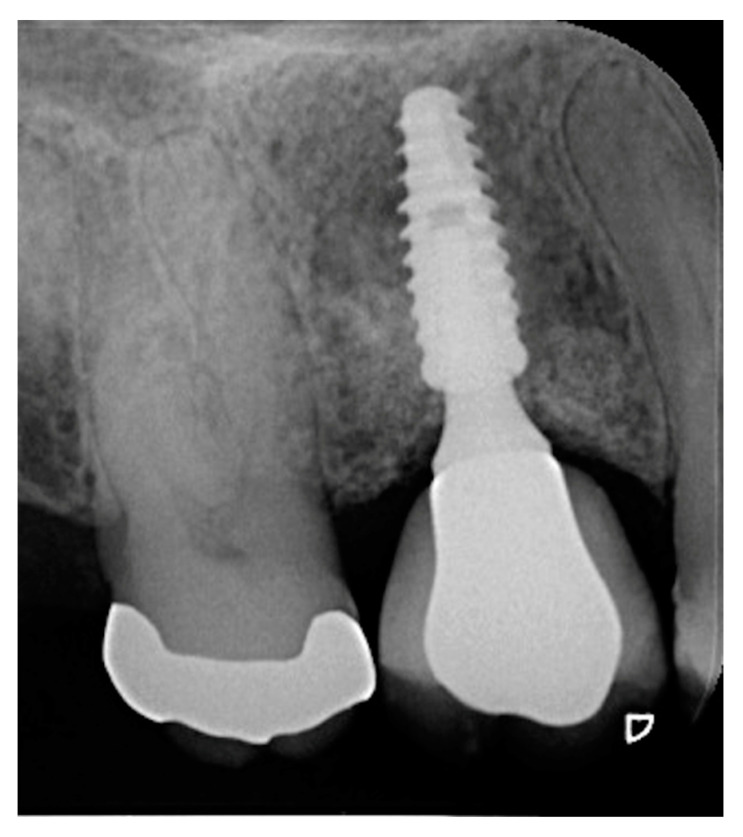
Periapical radiograph of #16 in function for 25 months.

**Table 1 materials-15-03539-t001:** Distribution of groups.

Maxilla	143 Implants
Mandible	68 Implants
Males (32–68 y/o, mean 54.3 y/o)	71 patients
Women (27–71 y/o, mean 49.8 y/o)	61 patients
Overall mean age	53.5 y/o
Mean follow-up time (12–32 months)	19 months
Narrow Implants (3.5)	84 Implants
Regular Implants (4.0)	99 Implants
Wide Implants (4.5; and 5.0)	28 Implants

**Table 2 materials-15-03539-t002:** Survival rates of dental implants according to subgroups.

Overall (207 implants)	98.1%
Mandible (67 implants)	98.5%
Maxilla (140 implants)	97.9%
Immediate loading (125 implants)	99.2%
Immediate loading on Maxilla (86 implants)	98.85%
Immediate loading on Mandible (39 implants)	100%
Narrow implants (82 implants)	97.6%
Narrow implants on immediate load (35 implants)	100%
Regular implants (98 implants)	98.9%
Regular implants on immediate load (71 implants)	98.6%
Wide implants (27 implants)	96.4%
Wide implants on immediate load (19 implants)	100%
Smokers (Light and Heavy) (27 implants)	100%

**Table 3 materials-15-03539-t003:** Distribution of the dental implants by region and load indications.

Immediate loading—maxilla	87 Implants
Delayed loading—maxilla	56 Implants
Immediate loading—mandible	39 Implants
Delayed loading—mandible	29 Implants
Single tooth	177 Implants
Full arch fixed rehabilitation	34 Implants
Anterior region	103 Implants
Posterior region	108 Implants

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
