# Peer review of "Effects of Osseodensification on Immediate Implant Placement: Retrospective Analysis of 211 Implants"

_materials, 2022, doi:10.3390/ma15103539_

Round 1

Reviewer 1 Report

The manuscript entitled “Effects of Osseodensification on Immediate Implant Placement: Retrospective Analysis of 211 Implants” submitted to Materials aims to evaluate and discuss results obtained on immediate implant placement with immediate and delayed loading protocols under Osseodensification bone instrumentation protocols.

The manuscript appears interesting with innovative insights into this method of implant site preparation.

I have some suggestions to improve deeply the quality of the manuscript, enriching the text with further notions.

  • English language: Minor spell check is required.
  • Abstract: Please, revise the text as in some parts it seems redundant and the English form is not always correct.
  • Introduction: this section needs some improvements.
    In addition to under-preparation of the implant site, I suggest referring to other methods of site preparation such as piezoelectric preparation, compaction with manual compactors and magnetodynamic preparation [PMID: 32937785 - PMID: 32098046].
  • Methods: I suggest to add the ethical committee number in the text part.
  • Why was the primary stability of the implants not measured?
  • In your opinion, could the use of different implant diameters and lengths in the study have influenced the results of the manuscript?I suggest that this notion be included in the limitations of the study.
  • Discussion: Like any surgical method, bone densification requires a learning curve. I suggest making this point at the beginning of the discussion by referring to how the skill of the operator can influence the type of implant site preparation [PMID: 32475099].
  • “Despite the limited long-term evidence of 213 success, Osseodensification has shown to improve primary stability of dental 214 implants[20,21,23].” I suggest referring to recent in vitro studies where it was analyzed the primary stability of implants placed following the osseodensification technique and bone compaction.
  • I suggest adding a fundamental part on the limitations of this study; furthermore, I suggest analysing the discrepancy between implant site preparation in the mandible and in the maxilla, performing the osteodensification technique (advantages and disadvantages).
  • Please, add an abbreviation section at the end of the manuscript.

After the suggested improvements, I think it is necessary to re-evaluate the text.

Author Response

Reviewer 1

1.Abstract: Please, revise the text as in some parts it seems redundant and the English form is not always correct.

Reply: we’ve made some text modifications to the English form, not only on the Abstract but on all the manuscripts by a native English-speaking colleague. Thank you for your suggestion;

2.Introduction: this section needs some improvements.
In addition to under-preparation of the implant site, I suggest referring to other methods of site preparation such as piezoelectric preparation, compaction with manual compactors and magnetodynamic preparation [PMID: 32937785 - PMID: 32098046].

Reply: the referred site preparation methods were included in the Introduction section. Thank you for the suggestion;

3.Methods: I suggest to add the ethical committee number in the text part.

Reply: the ethical committee number was added to the M&M section, as suggested;

4.Why was the primary stability of the implants not measured?

Reply: This retrospective study did not evaluate the ISQ. In addition,  ISQ is not a standard parameter used in Brazil.

  1. In your opinion, could the use of different implant diameters and lengths in the study have influenced the manuscript results? I suggest that this notion be included in the limitations of the study.

Reply: the idea was to analyze all implants placed with Osseodensification regarding the region, so an extensive range of implants diameters and lengths would have been installed for different situations. If we think about osseodensificaton theory, the larger the implant, the more trabecular bone should be condensed, and higher primary stability should be reached. But that is not what we found. The overall survival rate of narrow implants is more elevated than wide implants. But on immediate load cases, both achieved 100%. This data can be found in Table 2. We will include this discussion in the limitations of the study. Thank you for the suggestion;

  1. Discussion: Bone densification requires a learning curve like any surgical method. I suggest making this point at the beginning of the discussion by referring to how the operator's skill can influence the type of implant site preparation [PMID: 32475099].

Reply: thank you for the suggestion; we will include this information in the discussion section: “Clinician skills are an important factor that can influence the level of primary stability of the implants in different techniques [31], and must be similar to osseodensification burs. A periodontist placed all implants analyzed in this study with more than 18 years of experience”;

7.“Despite the limited long-term evidence of 213 success, Osseodensification has improved primary stability of dental 214 implants[20,21,23].” I suggest referring to recent in vitro studies analyzing the primary stability of implants placed following the osseodensification technique and bone compaction.

Reply: thank you for the suggestion. We included more in vitro studies to improve the discussion;” Recent in vitro studies compared Standard drilling sequences with osseodensification instrumentation protocols in low-density polyurethane blocks, and also concluded that OD resulted in higher primary stability values[ 31,32].”;

  1. I suggest adding a fundamental part on the limitations of this study; furthermore, I suggest analyzing the discrepancy between implant site preparation in the mandible and the maxilla, performing the osteodensification technique (advantages and disadvantages).

Reply: This information was added to the text: “When comparing survival rates achieved on the mandible, which usually has more cortical and dense bone, with maxillae, where we can find more trabecular bone to be osseodensificated, we didn’t find significant clinical differences, once we performed the instrumentation protocol suggested by the manufacturer (Versah), as can be seen in Table 2 also”;

  1. Please, add an abbreviation section at the end of the manuscript.

Reply: Thank you for the suggestion. We added an abbreviation section;

Reviewer 2 Report

The goal and novelty of work should be more described in introduction.

if the results can be shown in graph would be better.

the discussion is really weak, just repeating the results without any explanation and comparing with other studies.

Author Response

Reviewer 2

  1. The goal and novelty of the work should be described in the introduction.

Reply: we changed the way the objective of the manuscript was written, and now it may be more straightforward for you. “This retrospective study aimed to evaluate and discuss results obtained in 211 dental implants placed with Osseodensification bone instrumentation with the focused objective in immediate load cases, made by the same periodontist, from the beginning of his experience with this new technique.”

  1. If the results shown in the graph would be better.

Reply: Thank you for your suggestion. We will discuss this idea, as it would be necessary for many graphs to show all the results, and we may include some in the final revision;

  1. The discussion is feeble, repeating the results without any explanation and comparing with other studies.

Reply: we have made some improvements on the manuscript and also in the discussion, and now maybe satisfactory for you;  

Reviewer 3 Report

  1. The numbers of literatures in this retrospective study were insufficient to support the objective, the suggestion is that the retrospective literatures should be increased to enhance the value of the reference.
  2. The investigated parameters of the dental implant in the patients should be increased, such as survival rates correlated with age and bone density.
  3. The evaluation of statistical test for 132 subjects is suggested to investigate for providing additional information about survival rates by osseodensification bone instrumentation.
  4. The paragraph of the results must be described detail for better understanding.
  5. The paragraph of the discuss in length must be added to describe the correlation between the osseodensification bone instrumentation and the survival rate.

Author Response

  1. The numbers of literature in this retrospective study were insufficient to support the objective; the suggestion is that the retrospective literature should be increased to enhance the reference value.

Reply: During the review, we added several studies to support the objective and the results, and now we believe the manuscript has sufficient references;

  1. The investigated parameters of the dental implant in the patients should be increased, such as survival rates correlated with age and bone density.

Reply: Thank you for the suggestion. Some more information about the region of the implants( maxilla or mandible) correlating with bone density was added in the discussion; Age was not one of our goals, nor gender; “when comparing survival rates achieved on the mandible, which usually has more cortical and dense bone, with maxillae, where we can find more trabecular bone to be osseodensificated, we didn’t find significant clinical differences, once we performed the instrumentation protocol suggested by the manufacturer( Versah), as can be seen in Table 2 also.”

  1. The evaluation of a statistical test for 132 subjects is suggested to investigate to provide additional information about survival rates by osseodensification bone instrumentation.

Reply: “Statistical analysis of the implant survival rates was performed with life-table analysis as previously described by Mangano et al.,(2011)[27].”

  1. The paragraph of the results must be described in detail for better understanding.

Reply: The text changed a little to be more concise, and more information can be seen in Tables 1, 2, and 3;

  1. The paragraph of the length's discussion must be added to describe the correlation between the osseodensification bone instrumentation and the survival rate.

Reply: Some more information about the results was added to the discussion and maybe now satisfactory for you; Another limitation is that the implants studied had an extensive range of different lengths and diameters because of the other regions and situations in which they were installed. For that, the results could be different. But when we analyze Table 2, we can see that results on either narrow and wide implants were not so different, for example.”

Round 2

Reviewer 1 Report

Authors improved manuscript quality following the reviewer's suggestions.
I strongly recommend to check the format of references, avoiding possible grammatical errors and following the mendeley format.
It would also be helpful if the authors included other studies comparing osseodensification techniques with other implant site preparation methods.

Author Response

  1. I strongly recommend to check the format of references, avoiding possible grammatical errors and following the mendeley format.

Reply: Thank you for your suggestion. The manuscript was revised accordingly the Instructions for authors.

2.It would also be helpful if the authors included other studies comparing osseodensification techniques with other implant site preparation methods.

Reply: Well pointed. However, to date, there are few clinical articles evaluating the proposed technique. These papers were included int the revised manuscript. Please, see revised manuscript.

Reviewer 2 Report

after updating the references, this article can be accepted for publication in the journal.

Author Response

1.After updating the references, this article can be accepted for publication in the journal.

Reply: Thank you very much for the helpful thoughts.

Reviewer 3 Report

The limitations of the Osseodensification in patients could be descripted attentively in the discussion to increase a reference contribution.

The figure captions should be improved to explain more detail.

The latest literature about Osseodensification outcomes should be added, such as from 2021-2022.

Author Response

1.The limitations of the Osseodensification in patients could be descripted attentively in the discussion to increase a reference contribution.

Reply: Well pointed. The revised discussion section presents the limitations of both techniques and study design. In addition, the single failure demonstrated in our study probably could be consequence of bone density often observed in the mandibular area (more cortical than woven bone). Almutary et al.,(2018)[42] demonstrated that Osseodensification may not be effective in cortical bone, and it may function differently from trabecular bone, by decreasing bone repair and delaying or preventing osseointegration, in despite of increase primary stability of dental implants. This could be considered a limitation of Osseodensification technique, the need of at least 2mm of trabecular bone to be performed, and because of that, maybe it won’t be as effective in type I bone than in type III or IV”.

2.The figure captions should be improved to explain more detail.

Reply: Thanks. This version was revised accordingly.

3.The latest literature about Osseodensification outcomes should be added, such as from 2021-2022.

Reply: We included the last 2 papers published until May 5th, 2022.